# Cobalt Nanoparticles Embedded into N-Doped Carbon from Metal Organic Frameworks as Highly Active Electrocatalyst for Oxygen Evolution Reaction

**DOI:** 10.3390/polym11050828

**Published:** 2019-05-08

**Authors:** Jitao Lu, Yue Zeng, Xiaoxue Ma, Huiqin Wang, Linna Gao, Hua Zhong, Qingguo Meng

**Affiliations:** 1College of Chemical Engineering and Environmental Chemistry, Weifang University, Weifang 261061, China; lujitao@foxmail.com (J.L.); zengyue1998@foxmail.com (Y.Z.); maxiaoxue6666@126.com (X.M.); weifangwhq62@163.com (H.W.); 2College of Chemical and Environmental Engineering, Shandong University of Science and Technology, Qingdao 266590, China; gaolinna@sdust.edu.cn; 3Key Laboratory of Sensor Analysis of Tumor Marker, Ministry of Education, Shandong Key Laboratory of Biochemical Analysis, College of Chemistry and Molecular Engineering, Qingdao University of Science and Technology, Qingdao 266042, China; zhong82513@126.com

**Keywords:** metal-organic frameworks, N-doped carbon, oxygen evolution reaction

## Abstract

Cystosepiment-like cobalt nanoparticles@N-doped carbon composite named Co-NPs@NC with highly efficient electrocatalytic performance for oxygen evolution reaction was prepared from carbonization of N-doped Co-MOFs. The optimized Co-NPs@NC-600 shows overpotentials of 315 mV to afford a current density of 10 mA·cm^−2^. Meanwhile, the electrocatalys presents excellent long-term durability. The outstanding electrocatalytic performance can be attributed to the unique cystosepiment-like architecture with high specific surface area (214 m^2^/g), high conductivity of N-doped carbon and well-distributed active sites.

## 1. Introduction

Owing to ever-increasing fossil energy consumption and the corresponding environmental crisis, hydrogen energy has attracted increasing attention in the past few years due to its high energy density, recyclability and eco-friendly character [1,2,3,4,5,6,7,8]. Electrochemical water splitting is considered an efficient way to produce hydrogen through the half-reaction of the oxygen evolution reaction (OER) and hydrogen evolution reaction (HER) [9,10,11,12,13,14]. However, the process of electrochemical water splitting is severely plagued by the reaction kinetics of the OER at the cathode [15,16,17,18]. To date, noble metal catalysts based on platinum, ruthenium and iridium are the most efficient catalysts for the OER, but their high cost and scarcity render them impracticable for large scale industrial applications [19,20,21]. Therefore, it is highly desirable to develop highly efficient, low-cost and earth abundant electrocatalysts for OER [22,23,24,25,26].

To this end, considerable attempts have been devoted to developing transition metals and their compounds, such as nanosized transition metals, chalcogenides, hydroxides, layered double hydroxides, phosphides and pnictides [27,28,29,30,31,32]. Among them, nanosized transition metals, in particular metallic Co, have gained plenty of attention due to their inexpensive cost, earth abundance and high stability in alkaline solutions [33,34,35]. However, most of the reported nanosized transition metals show moderate electrocatalytic activity for OER, which is far from satisfactory. This is possibly because of their tendency to self-agglomerate, which severely restrict the quantity of surface active sites. An efficient way of surmounting these natural drawbacks is to fabricate nanohybrids with nanosized transition metals anchored in porous carbon materials. Moreover, this is also an effective method to protect the materials from the harsh environment and to ensure stability during the electrocatalytic processing [36].

Metal-organic frameworks (MOFs), a class of highly porous inorganic-organic hybrid materials constructed by metal ions/clusters and organic ligands, were considered to be excellent precursors to fabricate porous nanohybrids through thermolysis [37,38]. Transition metals@N-doped carbon composites with highly uniform composition, in particular, can be acquired using special MOFs containing a metal source, carbon source and nitrogen source as a self-sacrifice. It should be noted that N-doped carbon materials preserve abundant defects, which can improve the conductivity of carbon, resulting in the remarkable enhancement of their electrocatalytic performance. For example, Zhang et al. reported a highly efficient Co@N-doped carbon for both OER and HER using a Co-MOF as a self-sacrifice template [39]. Sun’s group fabricated a coral-like NiSe@N-doped carbon nanohybrid from the in-situ selenation of N-doped Ni-MOFs. This nanohybrid shows a highly efficient electrocatalytic performance for HER under all pH values [37]. Chen and co. prepared MOF derived CoSe_2_ nanoparticles embraced in N-doped graphitic carbon as an effective electrocatalyst for HER in an H_2_SO_4_ solution [40].

With those ideas in mind, cystosepiment-like cobalt nanoparticles @N-doped carbon composites named Co-NPs@NC were fabricated from the carbonization of N-doped Co-MOFs, {[Co(bcpb)(4,4′-bibp)_0.5_(H_2_O)_1.5_]·1.5H_2_O}_n_. The optimized Co-NPs@NC-600 shows a highly efficient electrocatalytic performance for the oxygen evolution reaction, which can be attributed to the unique cystosepiment-like architecture with a high specific surface area (214 m^2^/g), the high conductivity of N-doped carbon and the well-distributed active sites.

## 2. Experimental Section

### 2.1. Synthesis of Co-MOFs

Co-MOFs {[Co(bcpb)(4,4′-bibp)_0.5_(H_2_O)_1.5_]·1.5H_2_O}_n_ were synthesized according to the previous report [41]. In detail, a mixture of Co(NO_3_)_2_·6H_2_O (57 mg, 0.2 mmol), 4,6-di(4-carboxyphenyl)pyrimidine (0.2 mmol, 64 mg), 4,4′-di(1H-imidazol-1-yl)-1,1′biphenyl (57 mg, 0.2 mmol), NaOH (12 mg, 0.3 mmol) and H_2_O (12 mL) was heated at 170 °C for 3 days in a pressure-resistant Teflon-lined stainless container. Afterwards, pink crystals were collected by filtration, washed with fresh mother liquid and dried in air.

### 2.2. Synthesis of Co-NPs@NC-T (T = 500–700 °C)

The as-synthesized Co-MOFs that were put in a porcelain boat were annealed at 500, 600 and 700 °C for 3 h with a heating rate of 5 °C·min^−1^ under argon atmosphere. Then, Co-NPs@NC-T was obtained, where T represents the carbonization temperature. 

## 3. Results and Discussion

The Co-MOFs were synthesized by a solvothermal reaction of Co(NO_3_)_2_·6H_2_O, 4,6-di(4-carboxyphenyl)pyrimidine and 4,4′-di(1H-imidazol-1-yl)-1,1′biphenyl, as previously reported with some modifications. The X-ray diffraction (XRD) pattern of the Co-MOFs matches well with the simulated one of the single crystal, implying the phase purity of the compounds (Appendix A). Then, the as-prepared Co-MOFs are pyrolyzed to obtain cystosepiment-like Co-NPs@NC-T under N_2_ atmosphere at 600–700 °C. The phase of the Co-NPs@NC-T is confirmed by PXRD diffraction. As shown in Figure 1A, the peaks at 2θ = 44.22 and 51.52 can be assigned to the (111) and (200) planes of the face-centered-cubic (fcc) Co crystal (JCPDS No.15-0806). Moreover, no impurity peak is observed in Figure 1B, indicating the high phase purity of the samples. The nitrogen adsorption-desorption isotherm curve of the Co-NPs@NC-600 samples at 77 K is shown in Figure 1C, giving a type I of microporous adsorption at a small P/P_0_ and a hysteresis phenomenon during P/P_0_ = 0.4−1.0. Co-NPs@NC-600 presents a relatively large specific surface area of 214 m^2^/g, and the total pore volume is 0.229 cm^3^/g. The pore size distribution plot shows that the size of the mesoporous material is mainly in the range of 3−5 nm, generated during the pyrolysis process (Appendix A). Such structures can facilitate the mass/charge transportation and provide more catalytic active sites during the OER process. The components of the Co-NPs@NC-600 samples are further proved by Fourier transform infrared spectroscopy. As shown in Appendix A, the organic ligands are entirely removed for the Co-NPs@NC-600 samples. The Raman spectra of the NPs@NC-600 shows a D band at 1338 cm^−1^ (it arises from the sp^3^ defects), a G at 1584 cm^−1^ (it arises from the in-plane vibration of sp^2^-bonded pairs) and a 2D band at 2728 cm^−1^ (Figure 1D). The intensity ratio of the D and G bands is about 1.02, suggesting that plenty of defects exist in the carbon matrix, owing to the N-doping [42]. According to previous reports, the content of the N-doped carbon plays a key role in the electrocatalytic performance of the sample [41]. Hence, thermogravimetric analyses (TGA) were employed to determine the approximate contents of the N-doped carbon in Co-NPs@NC-T (Appendix A). Assuming that all of the metallic Co converted completely into Co_3_O_4_ under oxygen atmosphere, the N-doped carbon of Co-NPs@NC-600 is the biggest among the three samples, which shows that the carbonization temperature can affect the components of the Co-NPs@NC-T samples, thus enhancing the electrocatalytic performance. The morphology of Co-NPs@NC-T is examined by transmission electron microscopy (TEM) and scanning electronic microscopes (SEM). As shown in Figure 2A,B and Appendix A, the SEM of Co-NPs@NC-600 indicates that the formed Co nanoparticles are homogeneously embedded into the N-doped carbon architectures, resulting in the formation of cystosepiment-like nanocomposites. However, Co-NPs@NC-500 and 700 present an inhomogeneous and aggregation morphology, which could reduce the electrocatalytic activity during the OER process. According to the statistical analysis (Appendix A), the diameter of most of the Co nanoparticles vary from 10 to 25 nm, implying that the particles are quite uniform in their dimensions. The high-resolution TEM image (HR-TEM) shows two distinct lattice fringes with lattice spaces of 0.21 and 0.33 nm, corresponding to the (111) plane of the cubic cobalt crystal and the (001) plane of the carbon materials, respectively; this indicates the high crystallinity of the samples (Figure 2C). Scanning TEM energy dispersive X-ray spectroscopy (STEM-EDS) elemental mapping images of Co, Se and N for Co-NPs@NC-600 are shown in Figure 2D–F, indicating the uniform distribution of these elements on the samples. The surface composition and surface chemical states of Co-NPs@NC-600 are characterized by X-ray photoelectron spectroscopy (XPS). As depicted in Figure 3, the XPS analysis confirms the presence of Co, N, C and O. The presence of the O element may be attributed to the unavoidable adsorbed oxygen-species or the slight oxidation of the surface of the samples. The high resolution of C_1s_ can be fitted to two peaks (284.5 and 286.0 eV), which can be assigned to C–C bonds and C–N bonds. The peak at 286 eV confirms the presence of doping nitrogen [43]. The high resolution of N_1s_ can be deconvoluted into three peaks with binding energies at 397.8, 399.6 and 400.8 eV, which can be attributed to pyridinic N, pyrrolic N and graphitic N, respectively [44]. According to the previous reports, graphitic N and pyrrolic N can improve the conductivity of the samples, thereby promoting the electrocatalytic activity during the OER process [45]. After deconvolution, the high-resolution spectrum of Co_2p_ suggests the existence of Co^0^ (778.5 eV), Co^2+^ (795.1 and 780.5 eV), Co^3+^ (779.5 and 797.3 eV) and Co-Nx (782.5 eV), whereas oxidized Co may result from the oxidation of the metallic Co on the surface of the samples [9,46]. The XPS results indicate the successful synthesis of Co-NPs@NC-600.

The effect of the carbonization temperature on the OER performance is tested with linear sweep voltammetry (LSV) in 1 M KOH (Figure 4A). Obviously, Co-NPs@NC-600 shows the best OER activity in contrast to the other samples. As a comparison, blank Cu foam, commercial cobalt powder and a RuO_2_ counterpart are also investigated in the same conditions. As can be seen, the Cu foam barely presents any electrocatalytic performance for OER. The Co-NPs@NC-600 samples afford a current density of 10 mA·cm^−2^ at a small η of 310 mV, which is bigger than that of the state-of-the-art RuO_2_, but which is markedly smaller than that of commercial cobalt powder, previously reported MOF-derived cobalt and other outstanding Co-based OER catalysts (Appendix A). The Tafel slope of the electrode can be calculated through being fitted to the Tafel equation, which is an inherent property for estimating the electrocatalytic kinetics. As depicted in Figure 4B, the commercial RuO_2_ shows a low Tafel slope of 120 mV·dec^−1^. However, the Tafel slope of Co-NPs@NC-600 is 186 mV·dec^−1^, which is close to that of the commercial Co powder of 168 mV·dec^−1^, suggesting the same rate-determining steps [47]. The electrochemical impedance spectroscopy (EIS) technique is employed to investigate the electrode kinetics during the OER process from 100 MHz to 0.01 Hz at an overpotential of 315 mV (Figure 4C). According to the previous reports, the semicircles in the high- and low-frequency range of the Nyquist plot are associated with the charge-transfer resistance (*R*_ct_) and solution resistance (*R*_s_), respectively [48,49]. Both Co-NPs@NC-600 and commercial Co powder are fitted using an identical equivalent circuit: the charge-transfer resistance (*R*_ct_), solution resistance (*R*_s_), and constant-phase resistance (*R*_cp_) (Appendix A). Obviously, the Rct value of Co-NPs@NC-600 is smaller than that of commercial Co powder, indicating a superior activity of Co-NPs@NC-600 for OER, which may arise from the unique sponge-like architecture with a high specific surface area. To further shed light on the highly intrinsic OER activity of Co-NPs@NC-600, the electrochemically active surface area (ECSA) of Co-NPs@NC-600 and commercial Co powder are estimated from the calculation of the electrochemical double layer capacitances (Cdl) in the nonfaradaic region (Figure 4D–E and Appendix A). According to the previous reports, the ECSA is linearly proportional to its Cdl value. The values of Cdl for the samples can be calculated from the scan-rate dependent CV curves. Clearly, the Co-NPs@NC-600 samples shows a markedly larger Cdl value (51.72 mF·cm^−2^) than that of the commercial Co powder samples (20.70 mF·cm^−2^), indicating that Co-NPs@NC-600 has a larger ECSA than commercial Co powder for OER. Besides the OER activity, durability is another significant criterion for estimating a preeminent electrocatalyst. The durability of Co-NPs@NC-600 is evaluated by continuous cyclic voltammetry (CV) scanning between 1.2 and 1.7 V vs. RHE with a scan rate of 10 mV·s^−1^. As depicted in Figure 4F, the current density decreases very slightly after 2000 cycles. The high stability of Co-NPs@NC-600 during the OER process is attributed to the protected metallic Co, which is embedded in the porous carbon materials.

## 4. Conclusions

In summary, cystosepiment-like cobalt nanoparticles @N-doped carbon composites named Co-NPs@NC are successfully fabricated from the carbonization of N-doped Co-MOFs. The optimized Co-NPs@NC-600 shows a highly efficient electrocatalytic performance and excellent durability for OER, which can be attributed to the unique sponge-like architecture with a high specific surface area (214 m^2^/g), the high conductivity of the N-doped carbon and the well-distributed active sites.

## Figures and Tables

**Figure 1 polymers-11-00828-f001:**
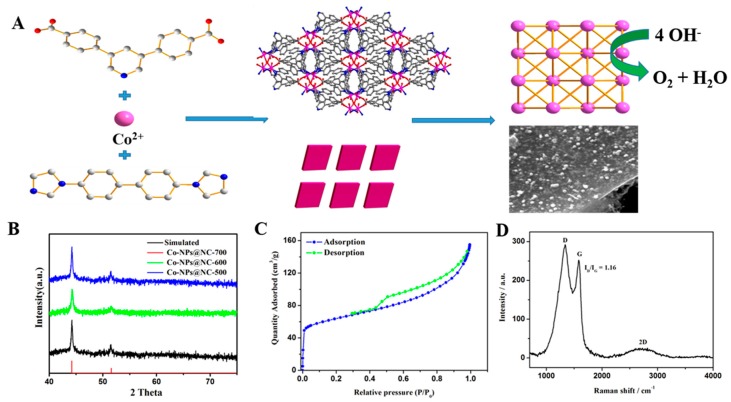
(**A**) Schematic illustration of the preparation process for the Co-NPs@NC-T samples from the Co-MOFs precursors, (**B**) the PXRD patterns of the Co-NPs@NC-T samples, (**C**) the N2 adsorption-desorption isotherms of the Co-NPs@NC-600 samples and (**D**) the Raman spectra of Co-NPs@NC-600 samples.

**Figure 2 polymers-11-00828-f002:**
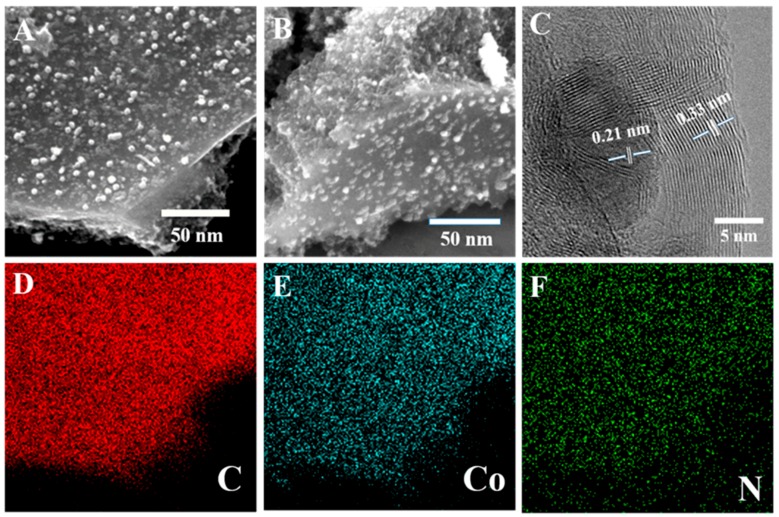
(**A**,**B**) SEM images of the Co-NPs@NC-600 samples, (**C**) HR-TEM images of Co-NPs@NC-600 samples, and (**D**–**F**) STEM-EDS elemental maps of Co-NPs@NC-600 samples.

**Figure 3 polymers-11-00828-f003:**
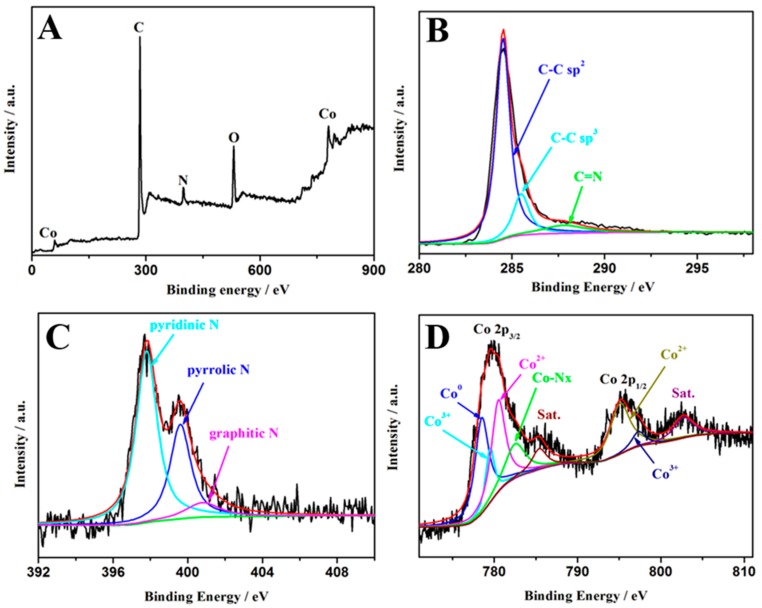
(**A**) Survey XPS spectrum, (**B**) high resolution of C 1s spectrum, (**C**) N 1s, (**D**) and Co 2p spectrum for the Co-NPs@NC-600 samples.

**Figure 4 polymers-11-00828-f004:**
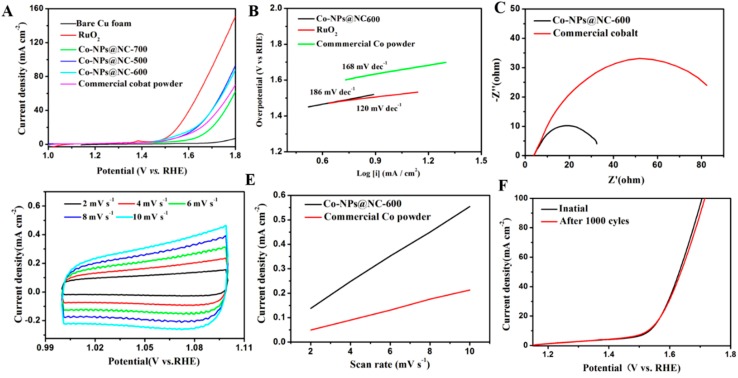
(**A**) Polarization curves of the Co-NPs@NC-T samples, benchmark RuO_2_, Cu foam and commercial Co powder in 1 M KOH, (**B**) polarization curve derived Tafel plots of the Co-NPs@NC-600 samples and commercial Co powder, (**C**) Nyquist plots of the Co-NPs@NC-600 samples and commercial Co powder, (**D**) Cyclic voltammograms at various scan rates of the Co-NPs@NC-600 samples, (**E**) Estimated Cdl of the Co-NPs@NC samples and commercial Co powder, and (**F**) polarization curves for the 1st and 1000th potential cycles of the Co-NPs@NC-600 samples in 1 M KOH.

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
