# Peer review of "Cobalt Nanoparticles Embedded into N-Doped Carbon from Metal Organic Frameworks as Highly Active Electrocatalyst for Oxygen Evolution Reaction"

_polymers, 2019, doi:10.3390/polym11050828_

Round 1

Reviewer 1 Report

The manuscript "Cobalt nanoparticles embedded into N-doped carbon from metal organic frameworks as highly active electrocatalyst for oxygen evolution reaction" by J. Lua, Y. Zenga, X. Maa, H. Wanga, L. Gaob and Q. Menga describes the synthesis of a novel catalyst as well as its characterization by several complementary experimental techniques. In general the manuscript is well written and the performed experiments are described accurately and concisely. The applied techniques are state of the art and the conclusions are sound. The topic of this article, a highly efficient catalyst for the production of molecular hydrogen, is of great importance and the article will surely be interesting for a broad readership.

Unfortunately, language and style do quite not meet the high scientific standards of this article. A thorough revision of the text by a native English speaker is therefore approved. Provided this has been done I recommend to publish this article.

In the following I give a short and surely incomplete list of putative typing errors:

Line 3: form (instead of from)

Line 14: from (instead of form)

Line 37: stability (instead of stable)

Line 38: show (instead of shows)

Line 40: nanohybrids (instead of nanohybirds)

Line 45: precursors (instead of precursor)

Line 54: It seems the hyperlink for ref. 33 is not working

Line 92: Fig. (instead of fig.)

Line 100: uniform (instead of uniformly)

Line 105: bonds (instead of bond)

Line 106: and (instead of and and)

Line 106: bonds (instead of bond)

Line 168: electrocatalytic (instead of electrtocatalytic)

Author Response

Reviewer 1:

The manuscript "Cobalt nanoparticles embedded into N-doped carbon from metal organic frameworks as highly active electrocatalyst for oxygen evolution reaction" by J. Lua, Y. Zenga, X. Maa, H. Wanga, L. Gaob and Q. Menga describes the synthesis of a novel catalyst as well as its characterization by several complementary experimental techniques. In general the manuscript is well written and the performed experiments are described accurately and concisely. The applied techniques are state of the art and the conclusions are sound. The topic of this article, a highly efficient catalyst for the production of molecular hydrogen, is of great importance and the article will surely be interesting for a broad readership.

Unfortunately, language and style do quite not meet the high scientific standards of this article. A thorough revision of the text by a native English speaker is therefore approved. Provided this has been done I recommend to publish this article.

In the following I give a short and surely incomplete list of putative typing errors:

1. Line 3: form (instead of from)

2. Line 14: from (instead of form)

3. Line 37: stability (instead of stable)

4. Line 38: show (instead of shows)

5. Line 40: nanohybrids (instead of nanohybirds)

6. Line 45: precursors (instead of precursor)

7. Line 54: It seems the hyperlink for ref. 33 is not working

8. Line 92: Fig. (instead of fig.)

9. Line 100: uniform (instead of uniformly)

10. Line 105: bonds (instead of bond)

11. Line 106: and (instead of and and)

12. Line 106: bonds (instead of bond)

13. Line 168: electrocatalytic (instead of electrtocatalytic)

Response: Thanks for the encouragement and constructive suggestion of this reviewer. We are sorry for the spelling mistakes in the manuscript. As can been seen in the revised manuscript, we have refined the language throughout the text carefully to optimize the English of this manuscript. Also, please find our response to the reviewer’s comments point by point below.

1. The title “Cobalt nanoparticles embedded into N-doped carbon from metal organic frameworks as highly active electrocatalyst for oxygen evolution reaction” have changed to “MOFs derived cobalt nanoparticles / N-doped carbon from metal organic frameworks as highly active electrocatalyst for oxygen evolution reaction”.

2. “form” have been replaced by “from”.

3. “stable” have been replaced by “stability”.

4. “shows” have been replaced by “show”.

5. “nanohybirds” have been replaced by “nanohybrids”.

6. “precursor” have been replaced by “precursors”.

7. We have corrected the mistake in the revised Ms.

8. We have revised the mistake.

9. “uniformly” have been instead of “uniform”.

10. “bond” have been replaced by “bonds”.

11. One of “and" have been delated.

12. The mistake have been revised.

13. “electrtocatalytic” have been changed to “electrocatalytic”

Reviewer 2 Report

This manuscript describes the synthesis, characterization, electrocatalytic property of N-dpoe carbon from Co-MOFs. The synthesis and characterization by PXRD, TEM, XPS spectrum, N2 gas-adsorption isotherm and electrocatalytic property for this compound are well described and can be considered to be accepted for the publication. About the N2 gas-ad/de-sorption isotherm of Co-NPs@NC-600 samples, the type I of microporous adsorption at small P/P0 and hysteresis phenomenon during P/P0 = 0.4 ~ 1.0 is observed. The explanation or description  is suggested to be added in the revised manuscript.

Author Response

This manuscript describes the synthesis, characterization, electrocatalytic property of N-dpoe carbon from Co-MOFs. The synthesis and characterization by PXRD, TEM, XPS spectrum, N2 gas-adsorption isotherm and electrocatalytic property for this compound are well described and can be considered to be accepted for the publication. About the N2 gas-ad/de-sorption isotherm of Co-NPs@NC-600 samples, the type I of microporous adsorption at small P/P0 and hysteresis phenomenon during P/P0 = 0.4 ~ 1.0 is observed. The explanation or description is suggested to be added in the revised manuscript.

Response: Thanks a lot for the encouragement and kind comments from this reviewer.

According to the constructive suggestion, “The nitrogen adsorption-desorption isotherm curve of Co-NPs@NC-600 samples at 77 K is shown in Fig. 1C, giving a type I of microporous adsorption at small P/P0 and hysteresis phenomenon during P/P= 0.4-1.0.” has been added in the revised manuscript.

Reviewer 3 Report

Manuscript Title: Cobalt nanoparticles embedded into N-doped carbon from metal organic frameworks as highly active electrocatalyst for oxygen evolution reaction

Manuscript ID: Polymers-459646

The manuscript is presents good research work and going to be quite interesting for readers and recommended for the publication after following minor corrections.    

1.  Authors need to incorporate some recent reference and reviews related to the subject to make it more interesting.

2. Author needs check manuscript thoroughly; specially for spelling mistakes and abbreviations (m2/gm, oC etc).

3. Authors need to include FT-IR data in the manuscript.

4. Authors need to provide pore size distribution and total pore volume data for the N-doped carbon samples and suitable explanation.

5. Please improve the quality of figure 5.

6. Author must include the comparison of electrocatalytic performance of their N-doped carbon samples with previously reported similar kind of materials in tabulated form.

7. Authors need to provide surface area data for all the samples (only for one sample is done) and compare them with related references.

8. Why only NC-600 sample shows best performance why not others; please provide proper explanation.

Author Response

The manuscript is presents good research work and going to be quite interesting for readers and recommended for the publication after following minor corrections.    

1.  Authors need to incorporate some recent reference and reviews related to the subject to make it more interesting.

2. Author needs check manuscript thoroughly; specially for spelling mistakes and abbreviations (m2/gm, oC etc).

3. Authors need to include FT-IR data in the manuscript.

4. Authors need to provide pore size distribution and total pore volume data for the N-doped carbon samples and suitable explanation.

5. Please improve the quality of figure 5.

6. Author must include the comparison of electrocatalytic performance of their N-doped carbon samples with previously reported similar kind of materials in tabulated form.

7. Authors need to provide surface area data for all the samples (only for one sample is done) and compare them with related references.

8. Why only NC-600 sample shows best performance why not others; please provide proper explanation.

Response: Thanks a lot for the comments and suggestions of this reviewer, all of which have been strictly followed during the process of revising the manuscript.

1. Thanks for the kind suggestion! As can be seen in the revised manuscript, some relevant referenceshave been cited as reference 7-4 and 13-14.

2. We are sorry for the spelling mistakes in the manuscript. As can been seen in the revised manuscript, we have corrected those mistake throughout the text carefully.

3. Good suggestion! FT- IR spectrum of Co-NPs@NC-600 has been added as Figure S3 and the corresponding discussion have been added in the revised manuscript.
4. Thanks for the constructive suggestion! The pore size distribution and total pore volume data for the N-doped carbon samples and suitable explanation have been added in the revised manuscript.

5. According to the kind suggestion, in the revised manuscript we redraw the Fig. 5 and now it seems good.

6. Some relevant references about similar kind of materials have been added in Table S1.

7. N2 adsorption and desorption isotherms of Co-NPs@NC-500 and Co-NPs@NC-700 have been tested for several times. Unfortunately, there is no N2 adsorption for Co-NPs@NC-500 samples. The N2 adsorption and desorption isotherms curve of Co-NPs@NC-700 is attached below.

8. Good suggestion! In the revised manuscript, TG, and SEM were employed to explain why NC-600 sample shows best performance.
